# Patient-Selection of a Clinical Trial Primary Outcome: The ENHANCE-AF Outcomes Survey

**Randall S. Stafford[1]\*, Eli N. Rice [2], Rushil Shah[3], Mellanie T. Hills[4], Julio C. Nunes[5], Katie DeSutter[3], Amy Lin[6], Karma Lhamo[2], Bryant Lin[7], Ying Lu[6], Paul J. Wang[3]**

**1** Stanford University School of Medicine, Stanford Prevention Research Center, Stanford, California, United States of America, **2** Stanford University School of Medicine, Center for Clinical Research, Stanford, California, United States of America, **3** Division of Cardiovascular Medicine, Stanford University School of Medicine, Stanford, California, United States of America, **4** Stop Afib.org, Greenwood, Texas, United States of America, **5** Department of Psychiatry, Yale University, New Haven, Connecticut, United States of America, **6** Department of Biomedical Data Science, Stanford University School of Medicine, Stanford, California, United States of America, **7** Division of Primary Care and Population Health, Stanford University School of Medicine, Stanford, California, United States of America

\* rstafford@stanford.edu

## Abstract

### Introduction

Before the initiation of the ENHANCE-AF clinical trial, which tested a novel digital shared decision-making tool to guide the use of anticoagulants in stroke prevention for patients with atrial fibrillation, this study aimed to identify the most appropriate, patient-selected primary outcome and to examine whether outcome selection varied by demographic and clinical characteristics.

### Methods

Our cross-sectional survey asked 100 participants with atrial fibrillation to rank two alternative scales based on the scales' ability to reflect their experiences with decision-making for anticoagulation. The Decisional Conflict Scale (DCS), a 16-item scale, measures perceptions of uncertainty in choosing options. The 5-item Decision Regret Scale (DRS) focuses on remorse after a healthcare decision. We included adults with non-valvular AFib and $CHA_2DS_2VASc$ scores of at least 2 for men and 3 for women. Multivariable logistic regression with backward selection identified characteristics independently associated with scale choice.

### Results

The DCS was chosen over the DRS by 77% [95% confidence interval (CI) 68 to 85%] of participants. All subgroups designated a preference for the DCS. Those with higher $CHA_2DS_2VASc$ scores (≥5, n = 26) selected the DCS 54% of the time compared with 86% of those with lower scores (<5, n = 74; $p = 0.002$). Multiple logistic regression confirmed a weaker preference for the DCS among those with higher $CHA_2DS_2VASc$ scores.

**Data availability statement:** All relevant data are available in the paper and its Supporting Information files.

**Funding:** Funding for the ENHANCE-AF project (NCT04096781), including this analysis was provided by the Joe and Linda Chlapaty Stanford DECIDE Center of the Atrial Fibrillation Strategically Focused Research Network Award, awarded by the American Heart Association (AHA, 18SFRN34120036 and 18SFRN34240003). The views, statements and opinions in this article are solely the responsibility of the authors and do not necessarily represent the official views of the AHA. The project was led by Stanford University, in partnership with East Carolina University, Ochsner Medical Center, Cooper Health Care/Rowan University, and the Cleveland Clinic in the United States. The AHA did not participate in the design, implementation, analysis or interpretation of the reported work.

**Competing interests:** The authors have declared that no competing interest exist.

**Non-Standard Abbreviations and Acronyms:** $CHA_2DS_2VASc$ Score, Estimation of stroke risk in atrial fibrillation composed of heart failure, hypertension, age, diabetes mellitus, stroke/transient ischemic attack, and vascular disease; ENHANCE-AF, Engaging Patients to Help Achieve Increased Patient Choice and Engagement for AF Stroke Prevention; DCS, Decisional Conflict Scale; DRS, Decision Regret Scale.

## Conclusions

Individuals with atrial fibrillation preferred the DCS over the DRS for measuring their decision-making experiences. As a result of this survey, the DCS was designated as the ENHANCE-AF clinical trial's primary endpoint.

## Introduction

Atrial fibrillation (AFib) is a common arrhythmia affecting more than 6 million individuals worldwide [1]. AFib is associated with significant morbidity, mortality, and socio-economic burden, particularly from stroke and thromboembolism [2]. Anticoagulation use, despite its potentially serious adverse effects, is beneficial for embolic stroke prevention in at-risk AFib patients. A significant proportion of eligible AFib patients, however, do not receive the full benefits of anticoagulation due to inadequate prescription practices and limited medication adherence and persistence, leading to suboptimal anticoagulation use and preventable embolic events [3,4]. Additionally, the process by which clinicians and patients discuss the initiation of anticoagulation is not standardized, and the level of patient participation in decision-making varies greatly [5]. Traditionally, the annual stroke risk calculator, $CHA_2DS_2VASc$ score computed by clinicians is used to recommend oral anticoagulation to all patients except those with minimal stroke risk [6]. However, this paternalistic decision-making process often occurs without engaging patient participation and can lead to a lack of patient understanding and satisfaction, wasted healthcare resources, and preventable adverse outcomes [7,8]. Shared decision-making strategies can, therefore, improve patient satisfaction, lead to better decisions, better align decisions with patient preferences, and increase medication adherence and persistence [8,9].

The ENHANCE-AF trial (Engaging Patients to Help Achieve Increased Patient Choice and Engagement for AF Stroke Prevention, NCT04096781) envisaged using patient-selected outcomes to evaluate the impact of our shared decision-making digital tool (details found at afibguide.com and afibguide.com/clinician) [8,9]. A patient-selected primary outcome would better assess the quality of decision-making in a way that is meaningful and helpful to those with AFib [7]. Patient-centered outcomes closely align with patient experiences and are associated with a better quality of life, symptom scores, and greater satisfaction with care [7]. In contrast, traditional approaches that use epidemiological outcomes to ascertain patient results are tied to standard diagnostic criteria, natural history, and pathophysiology, which may be inherently less important to patients. As a result, these outcomes often fail to capture the full spectrum of events that are most meaningful to patients. On the other hand, patient-centered outcomes direct attention to the intrinsic needs, beliefs, and emotions of patients in conjunction with a physician's medical expertise and assessment [7]. Over the past decade, there has been a notable shift in clinical research towards the inclusion of patient-centered outcomes. This manuscript reports on our effort to advance patient-centered outcome research by having patients with AFib determine the most appropriate outcome measure to evaluate the impact of a healthcare delivery system intervention. To our knowledge, no other major clinical trials have used patients' input for the selection of the primary outcome.

The ENHANCE-AF project sought to develop and rigorously test a novel digital patient-clinician shared decision-making tool focused on stroke prevention in patients with AFib [8]. The current study was carried out during the design phase of the digital tool [10] to determine the primary outcome measure for the subsequent clinical trial as well as to measure if patient-preferred outcomes are consistent across patient characteristics.

## Methods

This cross-sectional study using a survey was designed to have individuals with AFib determine the most appropriate outcome measure for the ENHANCE-AF clinical trial and ascertain if patient-preferred outcomes were consistent across patient demographic and clinical characteristics. Additionally, the study also helped to refine the eligibility, recruitment, and data management methods later used in the clinical trial. Notably, there was no overlap in participants between the two studies.

Although we had initially envisioned testing a broad range of alternative outcome measures, we determined that only two scales were suitable for direct comparison and assessment, the Decisional Conflict Scale (DCS) [11] and the Decision Regret Scale (DRS) [12]. Other scales considered included: Preparation for Decision-Making [13], Satisfaction with Decision scale [14], Multi-Dimensional Measure of Informed Choice [15], Observing Patient Involvement in Decision Making scale [16], and Combined Outcome Measure for Risk Communication and Treatment Decision Making Effectiveness [17]. These scales were excluded for several reasons, including lack of applicability to AFib, mismatch with the decision-making context, and overall participant burden.

The DCS is a comprehensive 16-item measurement tool that assesses various aspects of individuals' personal perceptions about past decisions such as information supplied, values clarity, support received, uncertainty, and effectiveness [11]. The DCS was based on the decisional conflict concept of Janis and Mann and later developed into a survey scale by the North American Nursing Diagnosis Association [11]. The 5-item DRS developed by the Ottawa Health Research Institute, Ottawa Hospital, ON, Canada measures several aspects of regret after a healthcare decision [12].

Between August 1, 2019, and November 7, 2019, we recruited an independent sample of 100 Stanford Healthcare AFib outpatients who satisfied the identical eligibility criteria established for our intended clinical trial [8]. Using a convenience sample from both cardiology and primary care clinics, we included adults with non-valvular AFib, $CHA_2DS_2VASc$ scores of at least 2 for men and 3 for women, and who were able to provide written consent and follow instructions in English or Spanish. We excluded participants with hemodynamically significant mitral stenosis, mechanical valve replacement, or other indications for anticoagulation, as well as those with contraindications to anticoagulation.

Each participant completed a 15-minute-long questionnaire consisting of seven sections and 34 questions (S1 Supplemental Material) administered in-person by a trained clinical research coordinator utilizing a computer-based Research Electronic Data Capture (REDCap) database. Prior to the study, the survey had been pretested in three AFib patients and received approval from our project's patient advocate with AFib. Individuals were given the choice to complete this survey in either English or Spanish. At survey initiation, participants were provided the DRS and DCS questions and asked to review, but not answer the questions. We ensured survey anonymity and confidentiality by implementing mechanisms to prevent unauthorized access. Participants ranked the DCS and DRS on the scales' ability to meaningfully reflect their "decision-making experience with atrial fibrillation stroke prevention and blood thinners." To facilitate the process, participants initially received the DCS and DRS forms and were then given ample time to comprehensively review and understand their content. Additionally, we gathered data on clinical diagnoses, duration of anticoagulation, and participant demographics. To foster patient engagement and empowerment, participants were also asked to select a favored clinical trial name among three that were proposed.

Human subjects institutional review board approval was obtained via an expedited review (Stanford IRB 51678). The study was conducted as per the original protocol. All study

participants completed the entire study without any interruptions. There were no missing data or protocol deviations. No adverse events were reported.

### Patient involvement

Beyond the outcomes survey that is the focus of this article, patient involvement was crucial at every stage of our project: from designing the digital tool and planning a clinical trial to conducting the randomized controlled trial and dissemination of the tool. Using an end-user design model, the digital Stanford Medicine AFib tool (afibguide.com and afibguide.com/clinician) was created to improve patient-centered outcomes. Refined around patient-identified needs, the tool was iteratively tested by 25 patients [10]. During clinical trial planning, we relied on our ENHANCE-AF patient advocate (MTH) to contribute to study design and serve as a bridge between the academic study team and the AFib community. After input from our patient advocate, we pretested our outcomes survey on three other AFib patients and incorporated their feedback prior to study recruitment. For the clinical trial (n = 1001) recruitment strategies were influenced by patient input. All trial participants exposed to the digital tool were asked to provide feedback about the tool [9]. Once completed, trial results and tool access were disseminated through scientific presentations, professional organizations, and the patient advocacy group, Stop Afib.org.

### Statistical analysis

We designed a survey to identify patient preferences for a primary outcome measure to be used in a clinical trial of the effectiveness of a digital shared decision-making tool [8, 9]. Our planned sample of 100 participants was selected to provide a 10% margin of error on the estimated proportion of the population favoring one scale over the other. That is, 40% of 100 participants preferring one scale and 60% preferring the other would yield a statistically significant difference at $p < 0.05$. The sample population was summarized using descriptive statistics: continuous variables were summarized as mean and standard deviations and categorical variables were listed as numbers and frequencies. We characterized $CHA_2DS_2VASc$ scores as median and interquartile range (IQR) given its right-skewed distribution of values. Two-tailed p-values below 0.05 were considered statistically significant. Multivariable logistic regression was used to identify demographic, concomitant comorbidities, and medication usage that statistically moderated the choice of outcome scale. Model selection was based on backward selection to minimize the Akaike Information Criterion. This approach allowed all potential predictors (see Table 1) to be entered into the model, followed by the sequential removal of those that did not significantly contribute to the model. Logistic regression assessed the independent multiplicative effect of these covariates on the odds of selecting the DCS over the DRS. R Studio (Boston, MA, USA) and STATA (Stata Corp., College Station, TX, USA) were utilized for statistical analysis.

### Results

One hundred participants with AFib were enrolled between August and November 2019. The mean participant age (SD) was 76 (9) years, and 22% of total participants were female (Table 1). Along with AFib, 72% of participants had hypertension, 36% had vascular disease, 27% had stroke, 24% had diabetes mellitus, and 18% had heart failure. The median $CHA_2DS_2VASc$ score was 3.5 (interquartile range 3-5, mean 3.7, SD 1.35), and 83% of participants were taking oral anticoagulants. The survey was completed by all participants (100%) in English. The co-morbidities, age, and anticoagulation use in the sample closely resembled that of individuals diagnosed with AFib in the United States [18] and the ENHANCE-AF clinical trial [9]. The sample for this survey study, however, had more participants who were male.

**Table 1.** Scale preference by participant characteristics with bivariate odds ratio for favoring the decision conflict scale over the decisional regret scale.

| | Decision Regret Scale Chosen | | Decisional Conflict Scale Chosen | | Odds Ratio (95% CI)* |
|---|---|---|---|---|---|
| | Number | Percentage | Number | Percentage | |
| Overall | 23 | 23% | 77 | 77% | |
| Age < 70 Years | 2 | 17% | 10 | 83% | 1.57 (0.32 to 7.73) |
| Age ≥ 70 Years | 21 | 24% | 67 | 76% | |
| Men | 18 | 23% | 60 | 77% | 0.98 (0.32 to 3.03) |
| Women | 5 | 23% | 17 | 77% | |
| Education ≤ 2-year degree** | 6 | 21% | 22 | 79% | 1.05 (0.35, 3.12) |
| Education ≥ 4-year degree | 13 | 21% | 50 | 79% | |
| Patients Currently Taking Oral Anti-Coagulants | 22 | 27% | 61 | 73% | 0.17 (0.02 to 1.38) |
| Patients Not Taking Oral Anti-Coagulants | 1 | 6% | 16 | 94% | |
| $CHA_2DS_2VASc$ Score ≥ 5 | 12 | 46% | 14 | 54% | **0.18 (0.07 to 0.51)** |
| $CHA_2DS_2VASc$ Score < 5 | 10 | 14% | 64 | 86% | |
| Patients with Hypertension | 20 | 28% | 52 | 72% | 0.31 (0.08 to 1.15) |
| No Hypertension | 3 | 11% | 25 | 89% | |
| Patients with Diabetes Mellitus | 9 | 38% | 15 | 62% | 0.38 (0.14 to 1.03) |
| No Diabetes Mellitus | 14 | 18% | 62 | 82% | |
| Patients with Heart Failure | 5 | 28% | 13 | 72% | 0.73 (0.23 to 2.32) |
| No Heart Failure | 18 | 22% | 64 | 78% | |
| Patients with Past Stroke | 9 | 33% | 18 | 67% | 0.47 (0.18 to 1.28) |
| No Past Stroke | 14 | 19% | 59 | 81% | |
| Patients with Vascular Disease | 8 | 22% | 28 | 78% | 1.07 (0.40 to 2.84) |
| No Vascular Disease | 15 | 23% | 49 | 77% | |

Abbreviations: 95% CI, 95% confidence interval; $CHA_2DS_2VASc$, score for estimation of stroke risk in atrial fibrillation composed of heart failure, hypertension, age, diabetes mellitus, stroke/transient ischemic attack, and vascular disease.

*<1.0 means the first listed characteristic favors the DRS over the DCS and > 1.0 means the first listed characteristic favors the DCS over the DRS relative to the overall mean propensity to favor the DCS.

** For Education, missing data were present for 4 participants favoring DCS and 5 favoring DRS.

Overall, 77% of participants [95% confidence interval (CI) 68 to 85%] ranked DCS above DRS. The bivariate comparisons showed that 86% of patients with low $CHA_2DS_2VASc$ scores (n = 74) selected DCS, compared to only 54% of those with high scores (n = 26) ($p$ = 0.002) (Table 1). We found no significant difference in the likelihood of favoring DCS based on other patient characteristics. Multiple logistic regressions confirmed that higher $CHA_2DS_2VASc$ scores were associated with a less strong preference for DCS (Table 2). The multivariable logistic regression predicted DCS vs. DRS preference (C-statistic 0.70) based on the inclusion of $CHA_2DS_2VASc$ score and anticoagulation status. In this model, predicted preference for DCS varied from 51% (high score on anticoagulation) to 96% (lower score not on anticoagulation). Of three compelling name choices, 47% of participants selected "ENHANCE-AF: Engaging Patients to Help Achieve Increased Patient Choice and Engagement for Atrial Fibrillation Stroke Prevention" as the preferred study name for the clinical trial.

## Discussion

This study evaluated AFib patient preference between two different decision-making outcomes scales - the Decisional Conflict Scale (DCS) [11] and the Decision Regret Scale (DRS) [12]. The 16-item DCS measures personal perceptions of uncertainty in choosing options. In contrast, the DRS primarily focuses on feelings of remorse following a healthcare decision.

**Table 2. Stepwise backward logistic regression model of predictors of DCS vs. DRS preference.**

| Patient characteristic | Odds ratio (95% CI)* | P-value |
|---|---|---|
| CHA$_2$DS$_2$VASc score ≥ 5 | **0.22 (0.08 to 0.62)** | **0.004** |
| Currently taking anticoagulants | 0.23 (0.011 to 1.23) | 0.16 |

C-statistic = 0.70

Abbreviations: 95% CI, 95% confidence interval.

*<1.0 this characteristic favors the DRS over the DSC relative to the overall mean propensity to favor the DCS.

CHA$_2$DS$_2$VASc: Score for estimation of stroke risk in atrial fibrillation composed of heart failure, hypertension, age, diabetes mellitus, stroke/transient ischemic attack, and vascular disease.

A majority (77%) of the participants chose the DCS over the DRS. Participant endorsement of the DCS over the DRS may reflect the disadvantages of the DRS noted by past researchers. In general, the DRS indicates minimal regret in many decision-making settings [19,20] and therefore may be less useful in differentiating the quality of decision-making. Other studies indicate lower consistency and internal reliability for the DRS compared to the DCS [21].

We evaluated whether some patient groups might vary in their scale preference. A high CHA$_2$DS$_2$VASc score was the only statistically significant moderator of scale choice. Participants with high scores (≥ 5) still preferred the DCS although much less strongly than those with lower scores (< 5). In addition, the multiple logistic regression modeling included oral anticoagulant consumption as a meaningful, but non-significant predictor of scale choice. Even the most extreme category formed by these two data elements continued to prefer the DCS over the DRS. On this basis, we designated the DCS as our primary outcome for our clinical trial with the DRS designated as a key secondary outcome variable. We also named our randomized controlled trial in accordance with the participants' preference.

## Study limitations

Our study has several potential limitations. Although we assessed patient preferences for the DCS vs the DRS, other potential patient-centered outcomes were not assessed. This study was conducted on those with AFib seen at Stanford Healthcare and may not be representative of all patients in the U.S. with AFib. The study findings are perhaps most pertinent to individuals with AFib who have access to tertiary healthcare facilities and possess health insurance.

## Conclusions

Individuals with AFib preferred the DCS (77%) over the DRS (23%) for measuring their decision-making experience with oral anticoagulation for stroke prevention. Except for CHA$_2$DS$_2$VASc scores, both bivariate and multivariate logistic regressions found no statistically significant correlates of scale choice, providing reassurance of patients' preference for the DCS across a range of patient characteristics. Our innovative strategy led to meaningful patient involvement in clinical trial design, thereby enhancing this trial's patient-centeredness. The subsequent ENHANCE-AF clinical trial set DCS as its primary endpoint. This trial demonstrated that use of our digital shared decision-making tool resulted in lower DCS scores when compared to usual care [9]. By engaging AFib patients in the clinical trial design process, we used a collaborative and innovative approach to choose an outcome measure meaningful to the target patient population. This approach of involving consumers with a clinical condition of interest in defining study outcomes has potential applications in other technically complex clinical scenarios. This study also speaks to the feasibility of gathering this useful input, rather

than having researchers impose this choice on their participants. Future randomized controlled trials should consider planned patient involvement in the selection of key trial outcomes.

## Supporting Information

**S1 File. ENHANCE-AF outcomes survey.**
(PDF)

## Author contributions

**Conceptualization:** Randall S. Stafford, Mellanie T. Hills, Bryant Lin, Paul J. Wang.

**Data curation:** Randall S. Stafford, Rushil Shah, Katie DeSutter.

**Formal analysis:** Randall S. Stafford, Amy Lin, Ying Lu, Paul J. Wang.

**Funding acquisition:** Randall S. Stafford, Paul J. Wang.

**Investigation:** Randall S. Stafford, Bryant Lin, Paul J. Wang.

**Methodology:** Randall S. Stafford, Mellanie T. Hills, Paul J. Wang.

**Project administration:** Katie DeSutter, Karma Lhamo.

**Writing – original draft:** Randall S. Stafford, Rushil Shah.

**Writing – review & editing:** Randall S. Stafford, Eli N. Rice, Julio C. Nunes, Paul J. Wang.

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
