## [Decision Letter · Decision Letter 0]

20 Oct 2024

PONE-D-24-40722Patient-Selection of a Clinical Trial Primary Outcome:

The ENHANCE-AF Outcomes SurveyPLOS ONE

Dear Dr. Rice,

Thank you for submitting your manuscript to PLOS ONE. After careful consideration, we feel that it has merit but does not fully meet PLOS ONE’s publication criteria as it currently stands. Therefore, we invite you to submit a revised version of the manuscript that addresses the points raised during the review process.

The work is well written and scientifically sound. The reviewers have provided detailed comments. I do not have Please revise the manuscript per their comments. 

We look forward to receiving your revised manuscript.

Kind regards,

Daniel Antwi-Amoabeng, MD, MSc.

Academic Editor

PLOS ONE

2. Please ensure you have included the registration number for the clinical trial referenced in the manuscript.

Additional Editor Comments:

I commend the authors for their commitment to include patient perspectives in the design of the trial. Such preliminary work should be the standard in trial with the potential to be practice-changing.

Reviewers' comments:

Reviewer's Responses to Questions

**Comments to the Author**

1. Is the manuscript technically sound, and do the data support the conclusions?

Reviewer #1: Yes

Reviewer #2: Yes

2. Has the statistical analysis been performed appropriately and rigorously? 

Reviewer #1: Yes

Reviewer #2: Yes

3. Have the authors made all data underlying the findings in their manuscript fully available?

Reviewer #1: No

Reviewer #2: Yes

4. Is the manuscript presented in an intelligible fashion and written in standard English?

Reviewer #1: Yes

Reviewer #2: Yes

5. Review Comments to the Author

Reviewer #1: I applaud the authors for putting together an effective manuscript detailing he pretrial selection of an appropriate clinical tool to assess patient outcomes and in assessing the patient clinical comorbidity burden ( CHADSVASC score) in relation to the preference of the particular questionnaire ( DCS vs DRS). I have some suggestions and recommendations as detailed below.

1. The authors describe a 15- minute long, seven sections and 34 questions long questionnaire: please include that as a supplement or a table to allow the readers an insight into assessment of patient preference.

2. Please describe/enlist components of the multivariate regression model utilized.

3. Please restructure the conclusion and discussion section: the last 6-7 lines of conclusion can be elaborated on in the discussion for the ease of the readers.

Reviewer #2: In the current paper, the authors aimed to find the most appropriate, patient-selected primary outcome of the ENHANCE-AF clinical trial and to examine whether outcome selection differed by demographical and clinical characteristics. Patients with AF selected the DCS (77%) over the DRS (23%) to measure their decision-making experience with OAC for stroke prevention. The authors decided on the DCS as the primary endpoint for the ENHANCE-AF clinical trial. Patient knowledge and compliance are essential for the treatment and follow-up of the patients (patient-centeredness). What about the impact of education on digital SDM tools to guide the use of OACs in stroke prevention for patients with AF?

Furthermore, how will the authors solve the problem of patients with previous strokes who have difficulty understanding the SDM tool? Patients with a history of stroke should be evaluated separately if they have had difficulty with such tools. Furthermore, you can revise your analysis using the CHADS-VA score rather than the CHADS-VASc score, consistent with the recent AF guidelines.

6. PLOS authors have the option to publish the peer review history of their article (what does this mean? ). If published, this will include your full peer review and any attached files.

**Do you want your identity to be public for this peer review?** For information about this choice, including consent withdrawal, please see our Privacy Policy .

Reviewer #1: No

Reviewer #2: **Yes: ** Ugur Canpolat, MD, Professor, Hacettepe University Faculty of Medicine, Department of Cardiology, Ankara, Turkey

---

## [Author Response · Author response to Decision Letter 0]

3 Dec 2024

Please note that the clinical trials number is available on page three (NCT04096781).

I have added the data as a supplemental file.

---

## [Decision Letter · Decision Letter 1]

23 Jan 2025

Patient-Selection of a Clinical Trial Primary Outcome:

The ENHANCE-AF Outcomes Survey

PONE-D-24-40722R1

Dear Dr. Rice,

We’re pleased to inform you that your manuscript has been judged scientifically suitable for publication and will be formally accepted for publication once it meets all outstanding technical requirements.

Kind regards,

Daniel Antwi-Amoabeng, MD, MSc

Academic Editor

PLOS ONE

Additional Editor Comments (optional):

Reviewers' comments:

Reviewer's Responses to Questions

**Comments to the Author**

1. If the authors have adequately addressed your comments raised in a previous round of review and you feel that this manuscript is now acceptable for publication, you may indicate that here to bypass the “Comments to the Author” section, enter your conflict of interest statement in the “Confidential to Editor” section, and submit your "Accept" recommendation.

Reviewer #2: All comments have been addressed

2. Is the manuscript technically sound, and do the data support the conclusions?

Reviewer #2: Yes

3. Has the statistical analysis been performed appropriately and rigorously? 

Reviewer #2: Yes

4. Have the authors made all data underlying the findings in their manuscript fully available?

Reviewer #2: Yes

5. Is the manuscript presented in an intelligible fashion and written in standard English?

Reviewer #2: Yes

6. Review Comments to the Author

Reviewer #2: The authors reasonably replied to all my previous comments. The paper has significantly improved, so I don't have any more comments.

7. PLOS authors have the option to publish the peer review history of their article (what does this mean? ). If published, this will include your full peer review and any attached files.

**Do you want your identity to be public for this peer review?** For information about this choice, including consent withdrawal, please see our Privacy Policy .

Reviewer #2: **Yes: ** Ugur Canpolat

---

## [Editor Report · Acceptance letter]

PONE-D-24-40722R1

PLOS ONE

Dear Dr. Rice,

I'm pleased to inform you that your manuscript has been deemed suitable for publication in PLOS ONE. Congratulations! Your manuscript is now being handed over to our production team.

Kind regards,

on behalf of

Dr. Daniel Antwi-Amoabeng

Academic Editor

PLOS ONE